# The Combination of Plant Extracts and Probiotics Improved Jejunal Barrier and Absorption Capacity of Weaned Piglets

Lijie Yang [1], Xiangming Ma [2], Chongwu Yang [3], Shan Jiang [1], Weiren Yang [1] and Shuzhen Jiang [1,*]

[1] Shandong Provincial Key Laboratory of Animal Biotechnology and Disease Control and Prevention, Department of Animal Sciences and Technology, Shandong Agricultural University, No. 61 Daizong Street, Tai'an 271018, China; b20193040351@cau.edu.cn (L.Y.); 2020110427@sdau.edu.cn (S.J.); wryang@sdau.edu.cn (W.Y.)

[2] Dongying Science and Technology Innovation Service Center, No. 359 Nanyi Road, Dongying District, Dongying 257091, China; 202110462@sdau.edu.cn

[3] Guelph Research and Development Center, Agriculture and Agri-Food Canada (AAFC), 93 Stone Road West, Guelph, ON N1G 5C9, Canada; chongwuyang66@gmail.com

* Correspondence: szjiang@sdau.edu.cn; Tel.: +86-186-5381-7377

**Abstract:** Plant extracts and probiotics play a vital role in maintaining animal intestinal health. However, their joint compatibility program still needs to be further explored. In our study, thirty two piglets (Duroc × Landrace × Yorkshire) were selected to divided into four treatments, which included basal diet, basal diet + 1000 mg/kg probiotics with added glucose oxidase (PGO), basal diet + 500 mg/kg *Illicium verum* extracts (IVE), and basal diet + 500 mg/kg IVE + 1000 mg/kg PGO. All the piglets were housed individually for the 42-d trial period after 7-d adaptation. Results showed that dietary supplementation of PGO and IVE increased the digestibility of ether extract (EE), crude protein (CP), and lysine ($p < 0.05$). Likewise, the net protein utilization (NPU) was also improved ($p < 0.05$). What is more, adding PGO and IVE reduced crypt depth, increased villus length, and chrionic gland ratio of piglets ($p < 0.05$). Additionally, IVE or PGO that was applied alone can increase the expression of Occludin, Zona occludens 1 (ZO-1), and Sodium-dependent glucose transporters 1 (SGLT1) in jejunum ($p < 0.05$). Our results strongly suggest that the combination of IVE and PGO can improve the nutrient digestibility of weaned piglets by increasing the expression of nutrient transport vectors (SGLT1 and CAT1) and tight junction proteins (Occludin and ZO-1) in the jejunum. In conclusion, the combination of plant extracts and probiotics is a vital strategy to improve animal health before the advent of antibiotic substitutes with absolute advantages.

**Keywords:** weaned piglets; *Illicium verum* extracts; jejunal barrier; nutrient digestibility; probiotics



## 1. Introduction

In recent years, the functions of probiotics and plant extracts, such as antibacterial, antioxidant, and maintaining intestinal health, have been gradually recognized by the public [1,2]. Probiotics are defined as living microorganisms that have beneficial effects on host health when ingested in sufficient quantities [3]. Studies have shown that different probiotics can play a positive role in the healthy development of animals [4–6], plants [7,8], and humans [9,10]. The first-generation probiotics are mainly used in the food industry, focusing on the health function of the body. The second-generation probiotics refer to the specific strains with sufficient dose that can help the body recover the healthy phenotype or can use the existing strains as the carrier of health promoting molecules, so they are also called live biotherapeutics [11]. At present, the probiotic market is mainly occupied by the first-generation probiotics. Before the advent of second-generation probiotics, the rational use and benefit maximization of existing probiotics has become an important issue for current researchers. The ways of probiotics regulating animal intestinal health mainly come from two aspects. Dominant flora theory: first, probiotics adhere to intestinal mucosal

epithelial cells by competing for nutrients and parasitic sites and inhibit the attachment and growth of harmful bacteria. Biological oxygen capture theory: secondly, when the spores of probiotics enter the intestinal tract, they accelerate the consumption of oxygen in the environment, ensure the anaerobic environment and the growth of dominant bacteria, and ultimately improve the production performance of the host [12–14]. *Illicium verum* extracts (IVE) is a volatile oil with a unique aromatic smell extracted from the fruits or leaves of *Illicium verum*, whose main components are trans anisole, anisaldehyde, and estragol [15,16]. The main functions of IVE are anti-bacterial, enhancing the body's antioxidant and immune functions [17,18]. Studies have shown that IVE can exert its antibacterial function by inhibiting the formation of biofilm and reducing bacterial adhesion [19]. In recent years, the studies on plant extracts and enzyme-added composite microecological preparations have mainly aimed at a single object [20,21], but there are few reports on the interaction between the two in different animals [22,23]. Therefore, we have reason to propose that on the premise that plant extracts play a bactericidal role, using microecological agents to reshape healthy intestinal flora and build a synergistic system will become an important problem to be solved by researchers. In our study, IVE and probiotics with added glucose oxidase were used as the subjects to analyze nutrient utilization, intestinal histomorphology, and expression of developing-related factors of weaned piglets to investigate the mechanism of their combined use in improving the body development.

## 2. Materials and Methods

### 2.1. IVE and Probiotics Supplemented Diet

IVE (*trans-anethole* $\geq$ 93.16% and *estragole* $\geq$ 1.16%) was obtained by steam distillation in the Chinese Herb Medicine Laboratory of Shandong Agricultural University (Tai'an, China). Probiotics with added glucose oxidase (PGO) containing *Bacillus subtilis* ($1.9 \times 10^{10}$ cfu/g), *Bacillus licheniformis PWD-1* ($2.0 \times 10^{10}$ cfu/g), *Lactobacillus* ($5.0 \times 10^{10}$ cfu/g), and glucose oxidase (3000 U/g) were provided by Beijing CRVAB Biotechnology Co., Ltd. (Beijing, China).

The basal diet was formulated according to the minimum nutrient requirements of piglets by the National Research Council (Table 1) [24]. Next, IVE and PGO were added proportionally to the basal diet for the preparation of experimental diets.

**Table 1.** Ingredients and nutrient levels of basal diet (air dry basis)%.

| Ingredients | Content (%) | Nutrients [2] | |
|---|---|---|---|
| Corn | 64.50 | Digestible Energy, MJ/kg | 13.81 |
| Whey powder | 5.00 | Crude Protein (%) | 19.82 |
| Soybean meal | 23.00 | Calcium (%) | 0.70 |
| Fish meal | 5.00 | Total Phosphorus (%) | 0.64 |
| L-Lysine HCl | 0.20 | Lysine (%) | 1.22 |
| CaHPO$_4$ | 0.70 | Sulfur Amino Acid (%) | 0.65 |
| Pulverized Limestone | 0.30 | Threonine (%) | 0.75 |
| NaCl | 0.30 | Tryptophan (%) | 0.22 |
| Premix [1] | 1.00 | | |
| Total | 100.0 | | |

[1] Supplied per kg of diet: VA 3300 IU, VD$_3$ 330 IU, VE 24 IU, VK$_3$ 0.75 mg, VB$_1$ 1.50 mg, VB$_2$ 5.25 mg, VB$_{12}$ 0.026 mg, pantothenic acid 15.00 mg, niacin 22.50 mg, biotin 0.075 mg, folic acid 0.45 mg, Mn (as MnSO$_4$·H$_2$O) 6.00 mg, Fe (as FeSO$_4$·H$_2$O) 150 mg, Zn (as ZnSO$_4$·H$_2$O) 150 mg, Cu (as CuSO$_4$·5H$_2$O) 9.00 mg, I (as KIO$_3$) 0.21 mg, Se (as Na$_2$SeO$_3$) 0.45 mg. [2] Digestible energy was the calculated value, and the other nutrient levels were analyzed value.

### 2.2. Preparation and Determination of IVE

Steam distillation method is adopted. Each batch uses 2 kg of *Illicium verum*. During the extraction process, the bottom temperature is ($96 \pm 3$) °C, the steam flow is ($700 \pm 50$) mL/h, and the steam pressure is ($0.025 \pm 0.01$) MPa. After obtaining low concentration oil–water mixture, IVE was obtained by concentration and separation through liquid separation funnel.

Determination of *trans-anethole* by gas chromatography. The chromatographic column was polyethylene glycol 20,000 (PEG-20M) at 70 °C for 1 min. Then, it was linearly heated to 220 °C at a rate of 2 °C/min. Finally, the temperature was kept at 220 °C for 20 min.

The inlet and detector temperature were 250 °C, the carrier gas flow rate was 1 mL/min, and the split ratio was 100:1. *Estragole* was determined by gas chromatography. The chromatographic column is methyl polysiloxane capillary column. The temperature of chromatographic furnace is 200 °C constant temperature. The temperature of injection port and detector is 250 °C. The flow rate of carrier gas is 30 mL/min and the split ratio is 1:75.

### 2.3. Animals, Experimental Design, and Management

Management and design of the experiment followed animal care rules approved by the Animal Nutrition Research Institute of Shandong Agricultural University and the Ministry of Agriculture of China for the Care and Use of Laboratory Animals (SDAUA-2020-0710). Thirty-two piglets (Duroc × Landrace × Yorkshire) at the age of 38 d with an average body weight of 14.96 ± 0.36 kg (mean ± SD) were selected and randomly divided into four treatments. Treatment 1 was fed with basal diet (IVE-PGO−), and treatments 2, 3, and 4 were fed the basal diet supplemented with 1000 mg/kg PGO (IVE-PGO+), 500 mg/kg IVE (IVE + PGO−), and 500 mg/kg IVE + 1000 mg/kg PGO (IVE + PGO+), respectively.

All piglets were kept in individual stainless-steel cages (0.48 m$^2$) at a piggery of Shandong Agricultural University during the entire experiment period. The piglets had free access to food and water during the entire experiment period. The experiment was conducted for 42 d following a 7-d adaptation.

### 2.4. Sample Collections for Analysis of Nutrient Availability

We collected and recorded the amount of feces and urine excreted by each pig every day. After mixing, the representative samples were stored at −20 °C. The samples of feed and feces were dried at 65 °C to constant weight, and the indexes were determined after 72 h of moisture regain. The analysis of dry matter (DM), organic matter (OM), ethyl ether extract (EE), and CP was according to AOAC [25]. The content of amino acids was determined by Shandong Hua'an Detection Technology Co., Ltd. with Hitachi 835 high-speed automatic amino acid analyzer (Tokyo, Japan).

### 2.5. Sampling for Determination of Jejunal Barrier and Absorption Capacity

Piglets were euthanized (slaughtered after being fainted by electric shock at the neck) after fasting for 12 h and the jejunum was immediately removed under sterile conditions on the last day of the experiment. One sample was then immediately frozen in liquid nitrogen and stored at −80 °C for the subsequent analysis of genes expression of Occludin, Zona occludens 1 (ZO-1), Sodium-dependent glucose transporters 1 (SGLT1), and Cationic amino acid transporter 1 (CAT1). Another sample was promptly fixed in Bouin's solution for 24 to 48 h for hematoxylin and eosin staining and immunohistochemical analysis.

### 2.6. Quantitative Real-Time PCR

Real-time Polymerase Chain Reaction (PCR) for cDNA templates was carried out using ABI 7500 Real Time PCR System (Applied Biosystems, Foster City, CA, USA). Total RNA was extracted preserved at −80 °C with Trizol kit (Invitrogen, Carlsbad, CA, USA). The quality of the total RNA samples was tested by spectrophotometry. The purity and concentration of RNA were determined using UV spectrophotometer (DS-11 Spectropho-tometer, DeNovix, Wilmington, DE, USA) at an absorbance ratio of 260/280 nm (values in the range 1.8 to 2.0 indicate a pure RNA sample). Specific primers targeting Occludin, ZO-1, SGLT1, CAT1, and glyceraldehyde-3-phosphate dehydrogenase (GAPDH) genes were designed based on pig gene sequences reported on GenBank using Primer 6.0 and synthesized by Shanghai bioengineering LTD (Table 2). Total RNA was reverse transcribed to cDNA using the Prime Script@ RT Master Mix Perfect Real Time Kit (DDR036A, TaKaRa, Dalian, China). A total volume of 20 μL of the PCR mixture and reagents were added in accordance with the fluorescent quantitative PCR kit instructions. The optimized qRT-PCR protocol included an initial denaturation step at 95 °C for 30 s, followed by 40 cycles at 95 °C for 5 s, 60 °C for 34 s, 95 °C for 15 s, 60 °C for 60 s, and a detection of fluorescence

signals at 60 °C. The amount of relative gene expression was expressed and calculated as being equal to $2^{-\Delta\Delta CT}$ [26].

**Table 2.** Primers sequences of PCR.

| Genes | Accession No. | Primer Sequence (5′ to 3′) | Product Size bp |
|---|---|---|---|
| GADPH | NM_001206359.1 | F: ATGGTGAAGGTCGGAGTGAA<br>R: CGTGGGTGGAATCATACTGG | 154 |
| SGLT1 | M34044 | F: ATGTACCTGTCTGTCCTGTCG<br>R: GTGTTGGAGATGGTCTTGGAG | 381 |
| CAT1 | NM_001012613 | F: CATCAAAAACTGGCAGCTCA<br>R: TGGTAGCGATGCAGTCAAAG | 185 |
| Occludin | U79554 | F:<br>TATGAGACAGACTACACAACTGGCGGCGAGTCC<br>R:<br>ATCATAGTCTCCAACCATCTTCTTGATGTG | 363 |
| ZO-1 | MX003353439.2 | F: CAGCCCGAGGCGTGTTTA<br>R: AAGGTGGGAGGATGCTGTTG | 150 |

### 2.7. Histological and Immunohistochemical Analysis

After dewaxing and hydration, the tissue sections were decolorized with xylene and alcohol, then stained with hematoxylin for 1 min, and then washed with water until the solution was colorless. They were then stained with eosin and observed under a microscope (Nikon ELIPSE 80i, Tokyo, Japan). At least 6 staining sections of 3 piglets were examined in each group.

As described by Yang et al. [26], sections were preincubated with microwave in sodium citrate buffer (0.01 mol/L, pH 6.0) for 20 min and then incubated in 10% hydrogen peroxide ($H_2O_2$) for 1.5 h to block endogenous peroxidase activity. To prevent nonspecific binding, sections were cultured in 10% normal goat serum for 1 h (ZSGB-BIO, Beijing, China). Then, the sections were incubated overnight with polyclonal rabbit antibodies SGLT1 (1:250, AB14685, Abcam, Cambridge, UK) and polyclonal rabbit antibodies ZO-1 (1:150, AB96587, Abcam, Cambridge, UK) at 4 °C. Then, the sections were immersed in diaminobenzidine tetrachloride (DAB kit, Tiangen PA110, Beijing, China) to detect immunostaining. Images were analyzed using Image Pro Plus 6.0 (Media Control Netics, Silver Spring, MD, USA) for integrated optical density (IOD).

### 2.8. Western Blotting

The total protein was isolated from jejunum and determined (Beyotime, Shanghai, China), and the protein content of each sample was adjusted to 60 μg. Samples were separated by polyacrylamide gel electrophoresis and then transferred to a fixed object-p transfer membrane (Solarbio, Beijing, China). After 2 h of incubation with 10% skim milk powder, they were washed three times with Tris (TBST, pH 7.6). All primary antibodies were incubated overnight. After washing with TBST, membranes were diluted with goat anti-rabbit IgG (1:5000, ab6721, Cambridge, UK) (1:5000, UK), then washed 5 times, soaked in high sensitivity luminescence reagent (Beijing, Shanghai, China) using fusion captain Advanced FX7 (Beijing Oriental Technology Development Co., Ltd., Beijing, China), and analyzed using IPP6.0.

### 2.9. Statistical Analysis

Data were analyzed by one-way ANOVA and double factor variance analysis using the SAS program (version 9.2, SAS Institute Inc., Cary, NC, USA). Significant differences among treatment means were separated by Duncan's multiple range tests. The data were initially analyzed using a completely randomized design. All statements of significance were based on the probability of $p < 0.05$.

## 3. Results

### 3.1. Nutrients Digestibility

Dietary supplementation of PGO and IVE increased the digestibility of EE and CP ($p < 0.05$, Table 3). Compared with the control group, the net protein utilization (NPU) of weaned piglets was increased by adding of PGO and IVE ($p < 0.05$). Pooled data showed that both IVE and PGO also improved the EE and CP digestibility of piglets ($p < 0.05$). Likewise, adding PGO and IVE also improved the NPU of piglets ($p < 0.05$). Significant interaction was observed between IVE and PGO in the digestibility of lysine ($p < 0.05$), but there is no interaction on digestibility of methionine, arginine, histidine, leucine, isoleucine, phenylalanine, threonine, valine, and total essential amino acids ($p > 0.05$, Table 4). Compared with control group, the lysine digestibility decreased in the order of (IVE + PGO+) > (IVE + PGO−) > (IVE-PGO+), and the tryptophan digestibility decreased in the order of (IVE-PGO+) > (IVE + PGO+) > (IVE + PGO−). Pooled data showed that adding PGO and IVE increased the digestibility of lysine ($p < 0.05$), while adding PGO increased the digestibility of tryptophan ($p < 0.05$).

**Table 3.** Effect of probiotics with added glucose oxidase (PGO) and *Illicium verum* extracts (IVE) on the nutrient digestibility of weaned piglets [1], %.

| Items | | Apparent Digestibility | | | | BV | NPU |
|---|---|---|---|---|---|---|---|
| | | DM | OM | EE | CP | | |
| IVE− | PGO− | 80.46 | 83.75 | 87.97 [c] | 76.30 [c] | 88.02 | 67.16 [b] |
| | PGO+ | 80.80 | 84.31 | 88.90 [b] | 77.92 [a] | 88.39 | 68.87 [a] |
| IVE+ | PGO− | 80.55 | 83.84 | 89.22 [ab] | 77.49 [b] | 88.87 | 68.87 [a] |
| | PGO+ | 80.87 | 84.33 | 89.69 [a] | 77.48 [b] | 89.00 | 68.96 [a] |
| SEM | | 0.201 | 0.309 | 0.180 | 0.397 | 0.527 | 0.546 |
| Pooled | IVE− | 80.63 | 84.03 | 88.44 [b] | 77.11 | 88.21 | 68.02 [b] |
| | IVE+ | 80.71 | 84.08 | 89.45 [a] | 77.49 | 88.94 | 68.91 [a] |
| | PGO− | 80.51 | 83.79 | 88.59 [b] | 76.90 [b] | 88.45 | 68.01 [b] |
| | PGO+ | 80.83 | 84.32 | 89.30 [a] | 77.70 [a] | 88.70 | 68.92 [a] |
| *p*-Value | IVE+ | 0.949 | 0.901 | 0.002 | 0.127 | 0.643 | 0.048 |
| | PGO+ | 0.519 | 0.227 | 0.044 | 0.043 | 0.327 | 0.026 |
| | PGO × IVE | 0.876 | 0.929 | 0.452 | 0.046 | 0.316 | 0.141 |

[1] Treatment 1 was fed with basal diet (IVE-PGO−), and treatments 2, 3, and 4 were fed the basal diet supplemented with 1000 mg/kg PGO (IVE-PGO+), 500 mg/kg IVE (IVE + PGO−), and 500 mg/kg IVE + 1000 mg/kg PGO (IVE + PGO+), respectively. In the same column, values with different small letter superscripts mean significant difference ($p < 0.05$). DM, dry matter; OM, organic matter; EE, ether extract; CP, crud protein; BV, biological value; NPU, net protein utilization.

**Table 4.** Effect of probiotics with added glucose oxidase (PGO) and *Illicium verum* extracts (IVE) on EAA digestibility of weaned piglets [1], %.

| Items | | Lysine | Methionine | Tryptophan | Arginine | Histidine | Leucine | Isoleucine | Phenylalanine | Threonine | Valine | TEAA |
|---|---|---|---|---|---|---|---|---|---|---|---|---|
| IVE− | PGO− | 83.94 [c] | 84.78 | 78.41 [c] | 88.24 | 86.53 | 79.55 | 77.34 | 78.52 | 78.25 | 77.97 | 81.65 |
| | PGO+ | 85.15 [b] | 83.84 | 81.95 [a] | 88.45 | 86.15 | 79.84 | 75.92 | 78.48 | 77.52 | 76.55 | 81.21 |
| IVE+ | PGO− | 85.21 [b] | 84.24 | 80.92 [b] | 88.31 | 86.11 | 79.45 | 77.75 | 79.15 | 77.43 | 76.58 | 81.63 |
| | PGO+ | 85.96 [a] | 85.72 | 81.07 [b] | 88.78 | 86.26 | 81.21 | 78.01 | 78.77 | 78.74 | 76.93 | 81.99 |
| SEM | | 0.23 | 0.41 | 0.36 | 0.25 | 0.33 | 0.46 | 0.47 | 0.40 | 0.43 | 0.44 | 0.24 |
| Pooled | IVE− | 84.55 [b] | 84.31 | 80.18 | 88.35 | 86.34 | 79.70 | 76.63 | 78.50 | 77.89 | 77.26 | 81.43 |
| | IVE+ | 85.59 [a] | 84.98 | 81.00 | 88.55 | 86.19 | 80.33 | 77.88 | 78.96 | 78.09 | 76.76 | 81.81 |
| | PGO− | 84.58 [b] | 84.51 | 79.67 [b] | 88.28 | 86.32 | 79.50 | 77.55 | 78.84 | 77.84 | 77.28 | 81.64 |
| | PGO+ | 85.56 [a] | 84.78 | 81.51 [a] | 88.62 | 86.21 | 80.53 | 76.97 | 78.63 | 78.13 | 76.74 | 81.60 |
| *p*-Value | IVE+ | 0.013 | 0.410 | 0.191 | 0.660 | 0.814 | 0.488 | 0.079 | 0.533 | 0.808 | 0.600 | 0.523 |
| | PGO+ | 0.017 | 0.829 | 0.009 | 0.614 | 0.866 | 0.269 | 0.418 | 0.756 | 0.681 | 0.594 | 0.942 |
| | PGO × IVE | 0.046 | 0.111 | 0.021 | 0.857 | 0.693 | 0.425 | 0.256 | 0.832 | 0.260 | 0.319 | 0.549 |

[1] Treatment 1 was fed with basal diet (IVE-PGO−), and treatments 2, 3, and 4 were fed the basal diet supplemented with 1000 mg/kg PGO (IVE-PGO+), 500 mg/kg IVE (IVE + PGO−), and 500 mg/kg IVE + 1000 mg/kg PGO (IVE + PGO+), respectively. EAA: Essential amino acid; TEAA: Total essential amino acid. In the same column, values with different small letter superscripts mean significant difference ($p < 0.05$).

No ocular pathological changes were found in jejunum tissue sections of all treatments (Figure 1). The villus length in addition of PGO group was higher than that of control group ($p < 0.05$, Table 5), and the chrionic gland ratio in addition of PGO and IVE group was higher than that of control group ($p < 0.05$). Moreover, the crypt depth in the group supplemented

with IVE was lower than that of other groups ($p < 0.05$). Significant interaction was observed between IVE and PGO in the chrionic gland ratio ($p < 0.05$). Pooled data showed that adding PGO and IVE reduced crypt depth and increased the villus length and chrionic gland ratio of piglets ($p < 0.05$).

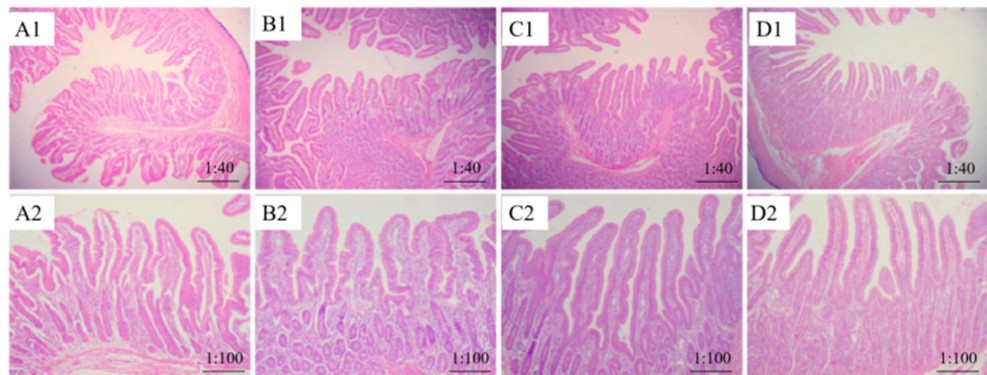

**Figure 1.** Effect of probiotics with added glucose oxidase (PGO) and *Illicium verum* extracts (IVE) on the jejunal histomorphology of weaned piglets. Treatment 1 was fed with basal diet (IVE-PGO−, (**A1,A2**)), and treatments 2, 3, and 4 were fed the basal diet supplemented with 1000 mg/kg PGO (IVE-PGO+, (**B1,B2**)), 500 mg/kg IVE (IVE + PGO−, (**C1,C2**)), and 500 mg/kg IVE + 1000 mg/kg PGO (IVE + PGO+, (**D1,D2**)), respectively.

**Table 5.** Effect of probiotics with added glucose oxidase (PGO) and *Illicium verum* extracts (IVE) on the jejunal villi and crypt depth of weaned piglets [1].

| Items | | Villus Length, μm | Crypt Depth, μm | Villus Length/Crypt Depth |
|---|---|---|---|---|
| IVE− | PGO− | 461.46 [b] | 390.17 [a] | 1.18 [b] |
| | PGO+ | 596.56 [a] | 333.80 [a] | 1.78 [a] |
| IVE+ | PGO− | 552.86 [ab] | 300.32 [b] | 1.84 [a] |
| | PGO+ | 551.61 [ab] | 324.91 [a] | 1.69 [ab] |
| SEM | | 14.461 | 12.312 | 0.093 |
| Pooled | IVE− | 529.02 | 361.99 | 1.48 |
| | IVE+ | 552.24 | 312.62 | 1.77 |
| | PGO− | 507.17 | 345.25 | 1.51 |
| | PGO+ | 574.09 | 329.36 | 1.74 |
| *p*-Value | IVE+ | 0.360 | 0.018 | 0.049 |
| | PGO+ | 0.034 | 0.896 | 0.033 |
| | PGO × IVE | 0.663 | 0.058 | 0.010 |

[1] Treatment 1 was fed with basal diet (IVE-PGO−), and treatments 2, 3, and 4 were fed the basal diet supplemented with 1000 mg/kg PGO (IVE-PGO+), 500 mg/kg IVE (IVE + PGO−), and 500 mg/kg IVE + 1000 mg/kg PGO (IVE + PGO+), respectively. In the same column, values with different small letter superscripts mean significant difference ($p < 0.05$).

### 3.2. Localization of SGLT1 and ZO-1

The immunopositive distribution of SGLT1 and ZO-1 in jejunum indicated that the expression of SGLT1 in IVE-PGO+, IVE + PGO−, and IVE + PGO+ groups was mainly distributed in the middle of jejunum villi, and gradually increased from the top to the middle (Figures 2 and 3). In addition, there were positive reactions in the villous chylous ducts. In the IVE-PGO− group, the positive reaction was strongest at the base of villi, but weaker at the upper and middle villi. The expression sites of ZO-1 are mainly distributed in the microvilli of jejunum and the wall of chylous villi, and the positive spots on the wall of chylous villi are mostly beaded.

### 3.3. The Expression of mRNA and Protein

The mRNA expression of ZO-1 and SGLT1 in jejunum of weaned piglets increased by adding IVE and PGO alone (Table 6, $p < 0.05$). In addition, PGO can also increase the mRNA expression of occludin and CAT1 ($p < 0.05$). The combination of IVE and PGO had significant interaction on increasing the mRNA expression of ZO-1 and SGLT1 in the jejunum ($p < 0.05$) and also showed a trend of promoting the mRNA expression of CAT1 ($p < 0.10$).

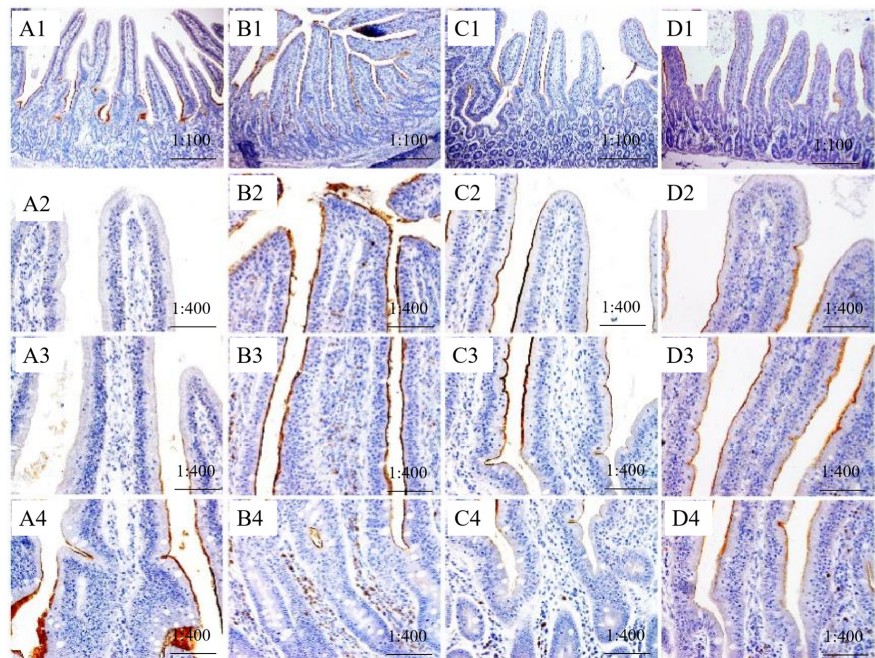

**Figure 2.** Effect of probiotics with added glucose oxidase (PGO) and *Illicium verum* extracts (IVE) on the immune positive distribution of Sodium/Glucose Cotransporter 1 (SGLT1) in jejunum of weaned piglets. Treatment 1 was fed with basal diet (IVE-PGO−, (**A1**–**A4**)), and treatments 2, 3, and 4 were fed the basal diet supplemented with 1000 mg/kg PGO (IVE-PGO+, (**B1**–**B4**)), 500 mg/kg IVE (IVE + PGO−, (**C1**–**C4**)), and 500 mg/kg IVE + 1000 mg/kg PGO (IVE + PGO+, (**D1**–**D4**)), respectively.

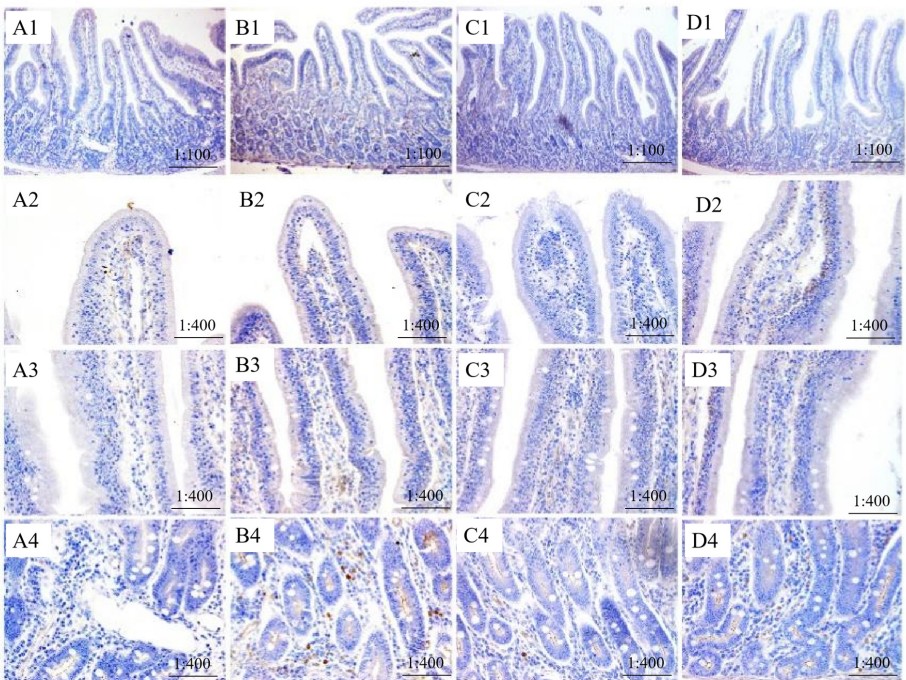

**Figure 3.** Effect of probiotics with added glucose oxidase (PGO) and *Illicium verum* extracts (IVE) on the immune positive distribution of Zona Occludens 1 (ZO-1) in jejunum of weaned piglets. Treatment 1 was fed with basal diet (IVE-PGO−, (**A1**–**A4**)), and treatments 2, 3, and 4 were fed the basal diet supplemented with 1000 mg/kg PGO (IVE-PGO+, (**B1**–**B4**)), 500 mg/kg IVE (IVE + PGO−, (**C1**–**C4**)), and 500 mg/kg IVE + 1000 mg/kg PGO (IVE + PGO+, (**D1**–**D4**)), respectively.

**Table 6.** Effect of probiotics with added glucose oxidase (PGO) and *Illicium verum* extracts (IVE) on the relative mRNA expression of jejunum in weaned piglets [1].

| | Items | Occludin | ZO-1 | SGLT1 | CAT1 |
|---|---|---|---|---|---|
| IVE− | PGO− | 1.00 [b] | 1.00 [c] | 1.00 [b] | 1.00 [b] |
| | PGO+ | 1.48 [a] | 2.30 [a] | 1.36 [a] | 1.83 [a] |
| IVE+ | PGO− | 1.28 [ab] | 1.57 [b] | 1.41 [a] | 1.24 [ab] |
| | PGO+ | 1.48 [a] | 1.92 [a] | 1.35 [a] | 1.63 [ab] |
| SEM | | 0.201 | 0.288 | 0.177 | 0.314 |
| | IVE− | 1.24 | 1.65 | 1.18 | 1.42 |
| Pooled | IVE+ | 1.28 | 1.75 | 1.38 | 1.44 |
| | PGO− | 1.04 | 1.29 | 1.21 | 1.12 |
| | PGO+ | 1.48 | 2.11 | 1.36 | 1.73 |
| | IVE+ | 0.704 | 0.071 | 0.032 | 0.165 |
| *p*-value | PGO+ | 0.010 | <0.001 | 0.002 | <0.001 |
| | PGO × IVE | 0.266 | 0.04 | 0.013 | 0.093 |

[1] Treatment 1 was fed with basal diet (IVE-PGO−), and treatments 2, 3, and 4 were fed the basal diet supplemented with 1000 mg/kg PGO (IVE-PGO+), 500 mg/kg IVE (IVE + PGO−), and 500 mg/kg IVE + 1000 mg/kg PGO (IVE + PGO+), respectively. ZO-1, Tight junction protein 1 (Zona occludens 1); SGLT1, Sodium-dependent glucose transporters 1; CAT1, Recombinant Cationic Amino Acid Transporter 1. SEM: standard error of the means; Values with a column with the different letters mean significantly different ($p < 0.05$).

The results of western-blot showed that dietary supplemented with IVE or PGO alone can increase the protein expression of jejunal Occludin, ZO-1, and SGLT1 in weaned piglets (Figure 4, $p < 0.05$). In addition, IVE can also increase the protein expression of CAT1 ($p < 0.05$). PGO and IVE showed significant interaction on the protein expression of Occludin, ZO-1, SGLT1, and CAT1 in jejunum of weaned piglets ($p < 0.05$).

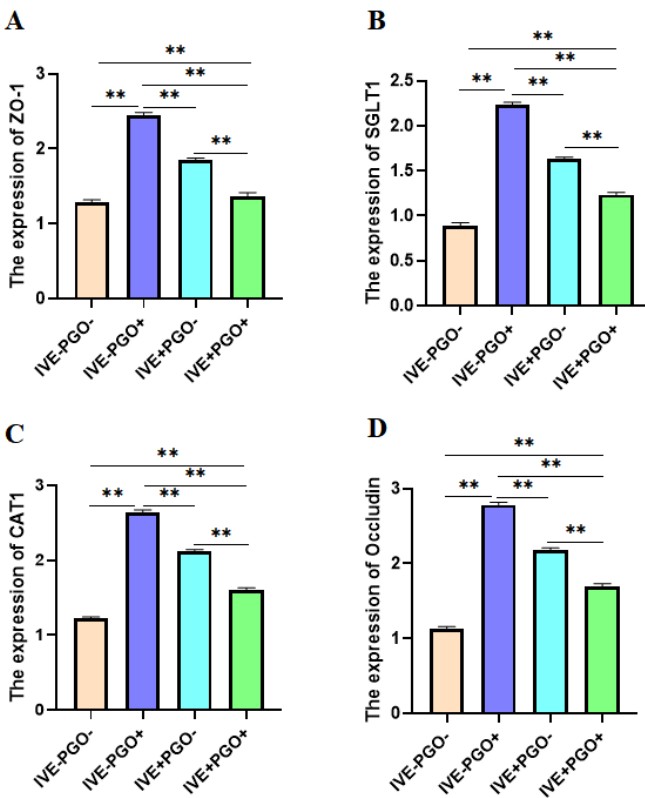

**Figure 4.** Effect of probiotics with added glucose oxidase (PGO) and *Illicium verum* extracts (IVE) on western blot of Zona Occludens 1 (ZO-1), Sodium/Glucose Cotransporter 1 (SGLT1), Cationic Amino Acid Transporter 1 (CAT1), and Occludin in jejunum of weaned piglets. Treatment 1 was fed with basal diet (IVE-PGO−), and treatments 2, 3, and 4 were fed the basal diet supplemented with 1000 mg/kg PGO (IVE-PGO+), 500 mg/kg IVE (IVE + PGO−), and 500 mg/kg IVE + 1000 mg/kg PGO (IVE + PGO+), respectively. (**A–D**) represents the expression on western blot of ZO-1, SGLT1, CAT1, and Occludin in jejunum. ** $p < 0.01$.

## 4. Discussion

### 4.1. Nutrients Digestibility

In our study, strains which can coexist with IVE were selected, and they established a synergistic system after compatibility. What is more, it was combined with IVE in the diet of weaned piglets. Although there was no statistical difference in growth performance, it played a positive role in improving nutrients digestibility. Studies have shown that the addition of compound probiotics composed of *Lactobacillus*, *Bacillus subtilis*, *Yeast*, and *Bifidobacteria* in the diet can improve the apparent digestibility of CP, EE, crude fiber (CF), and calcium (Ca) of weaned piglets [27,28]. This may be because the secondary metabolites of microorganisms contain some substances that can promote intestinal development. However, the study of IVE in weaned piglets is rare. Chen et al., (2017) showed that the addition of IVE did not affect the apparent digestibility of DM, OM, and CP in weaned piglets [29]. In our study, the supplementation of IVE and probiotics with added glucose oxidase can promote the digestibility of EE and CP, however, they only interact with the digestibility of CP. In our strategy, IVE reduces the abundance of harmful bacteria by inhibiting the formation of bacterial biofilm. In addition, when IVE is used in combination with probiotics, probiotics and harmful bacteria competitively adhere to intestinal mucosal epithelial cells and compete with pathogenic microorganisms for nutrients and ecological sites, therefore enhancing the biological barrier function of animal intestines and helping the host establish a healthy intestinal flora. This also improves the digestibility and growth performance of nutrients. In addition, we found that both IVE and probiotics with added glucose oxidase could improve the digestibility of amino acid, especially lysine, but no synergistic effect was observed. Similar conclusions have been confirmed in broiler chickens [30] and laying hens [31].

### 4.2. Jejunum Histomorphology

The integrity of intestinal villi and intestinal wall in jejunum is one of the important factors affecting the digestion also absorption of small intestine, and the ratio of villus length and crypt depth can reflect the basic relationship between intestinal villus and crypt [6,32]. Studies showed that plant extracts such as sugarcane [33] and red pepper [34] could improve the villus morphology and villus length/crypt depth. In recent years, the positive effect of probiotics on intestinal morphology has been recognized by researchers. Another study showed that when *Bacteriophage*, *Lactobacillus Acidophilus*, *Bacillus Subtilis*, and *Saccharomyces Cerevisiae* were combined to form a composite microecological agent, this improved the ratio of villus length and crypt depth [35]. In addition, in the study of Mishra et al., (2016), *Saccharomyces Cerevisiae NCDC-49* and *Lactobacillus Acidophilus-15* were combined to feed weaned piglets, which also improved the ratio of villus length and crypt depth [36]. What is more, we also observed the positive effect of the combination of PGO and IVE preparation on the ratio of villus length and crypt depth. Similarly, conclusions were also confirmed by Dowarah et al., (2017) [37]. By observing and comparing the jejunal sections of the four treatment groups, we found that the jejunal villi of piglets supplemented with PGO showed better morphology. It is worth noting that the combination of PGO and IVE has no superposition effect on the improvement of jejunal villus morphology. Although the villus morphology was also appropriately improved in the IVE + PGO + group, the ratio of villus length and crypt depth did not change significantly. Therefore, we believe that PGO and IVE play their prebiotic role by acting on different targets.

### 4.3. Localization and Expression

SGLT1, as a membrane transport vector, is mainly expressed on the surface of villi. In our study, the positive reaction of SGLT1 increased gradually from the top to the middle of villi. Additionally, some positive reaction points in the chylous duct of villi were detected, which were scattered. Similar conclusions have been reached in the study of Yoshida et al., (1995) and Moran et al., (2010) [38,39]. In addition, zonula occludens-1 (ZO-1) is one of the components of tight junction, which is mainly distributed in cell space and chyle tube wall. SGLT1 is a membrane transporter for glucose transport on the surface of small intestinal

villi and plays a vital role in glucose uptake [39]. In our study, IVE and PGO were found to promote the expression of SGLT1 in small intestinal villi of weaned piglets. Similar conclusions were confirmed in the study of Silveira et al., (2018) [40]. The difference of strain composition is an important reason for the inconsistent expression of SGLT1. Another result showed that the expression of ZO-1 was significantly increased by feeding probiotics, consisting of 8 strains of *Lactobacillus acidophilus*, *Lactobacillus bulgaricus*, and *Lactobacillus casei* [41]. However, the opposite conclusion was also observed [42]. The expression of ZO-1 is closely related to intestinal permeability. Inoue et al., (2006) found that the increase of the expression of tight junction proteins such as ZO-1 and occludin is synchronized with the recovery of intestinal barrier function [43]. Plant extracts such as *Quercetin* [44], *Pseudomonas* [45], and probiotics such as *Lactobacillus plantarum* [41], *Lactobacillus suis* [46], *Lactobacillus rhamnosus* [47], and *Bacillus* [48] can upregulate the expression levels of ZO-1 and occludin. In this study, *trans anethole*, *anisaldehyde*, *estragole*, and other plant active components in IVE may promote the expression of tight junction protein. It is suggested that PGO and IVE can improve the expression of tight junction protein in jejunal epithelial cells of piglets, improve jejunal permeability, and enhance its barrier function.

As a basic amino acid transporter, CAT1 plays a vital role in the absorption of lysine, arginine, and histidine [49]. At present, the research on CAT1 mainly focuses on lysine. Studies showed that there is a positive correlation between CAT1 and lysine in porcine small intestinal epithelial cells [50]. Another study showed that apple polyphenols could increase the expression of CAT1 in the muscle tissue of finishing pigs [51]. In our study, we found that IVE upregulated the expression of CAT1, but the effect of PGO on CAT1 was not observed. Therefore, we speculate that IVE can further improve lysine digestibility of piglets by increasing CAT1 expression. In addition, although PGO improved the digestibility of lysine, the expression of CAT1 did not change, suggesting that PGO may improve the absorption of lysine by regulating it in other ways, and relevant studies need to be further confirmed.

It must be mentioned that the difficult colonization of probiotics in the host intestine is a huge obstacle to the promotion and application of the first-generation probiotics. Studies shows that in the use of most probiotics, they only act as a passing bacterium [52]. However, studies have also shown that probiotics do not necessarily play a role only when they are alive [53,54]. Extracellular enzymes and ester peptides produced by probiotics when they are alive also play an important role in improving host health [55,56]. However, we did perform 16S rRNA sequencing analysis on animal feces in the early stages of the experiment and found that the abundance of some beneficial bacteria such as *Christensen* increased, but we did not find that the abundance of the genus of probiotics we invested in increased. Therefore, we speculate that the probiotics used under the test conditions may play a role through its secondary metabolites.

## 5. Conclusions

Our results strongly suggest that the combination of IVE and PGO can improve the nutrient digestibility of weaned piglets by increasing the expression of nutrient transport vectors (SGLT1 and CAT1) and tight junction proteins (occludin and ZO-1) in the jejunum. Under the global trend of limiting the use of antibiotics, our research provides a new idea for the selection of antibiotic substitutes. At present, among the most alternative antibiotics, the first-generation probiotics are widely used, but the effect is not ideal when used alone. It is suggested that the combination of plant extracts and probiotics is a vital strategy to improve animal health before the advent of antibiotic substitutes with absolute advantages.

**Author Contributions:** Conceptualization, L.Y. and X.M.; methodology, C.Y.; software, S.J. (Shan Jiang); validation, L.Y. and S.J. (Shan Jiang); formal analysis, X.M. and S.J. (Shan Jiang); investigation, S.J. (Shan Jiang); resources, W.Y.; data curation, S.J. (Shan Jiang); writing—original draft preparation, L.Y.; writing—review and editing, W.Y. and S.J. (Shuzhen Jiang); visualization, C.Y.; supervision, X.M.; project administration, S.J. (Shuzhen Jiang) and W.Y.; funding acquisition, S.J. (Shuzhen Jiang). All authors have read and agreed to the published version of the manuscript.

**Funding:** This research was funded by Shandong Province Pig Industry Technology System, grant number SDAIT-08-05, and Major Innovative Projects of Shandong Province, grant number 2019JZZY020609.

**Institutional Review Board Statement:** The study was conducted according to the guidelines of the Declaration of Helsinki and approved by the Animal Nutrition Research Institute of Shandong Agricultural University and the Ministry of Agriculture of China for the Care and Use of Laboratory Animals (SDAUA-2020-0710).

**Informed Consent Statement:** Not applicable.

**Data Availability Statement:** Not applicable.

**Conflicts of Interest:** The authors declared that they have no conflicts of interest to this work. We declare that we do not have any commercial or associative interest that represents a conflict of interest in connection with the work submitted.

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
