# Peer review of "The Combination of Plant Extracts and Probiotics Improved Jejunal Barrier and Absorption Capacity of Weaned Piglets"

_agriculture, doi:10.3390/agriculture12070912_

Round 1

Reviewer 1 Report

·       Abbreviations should be defined starting from the abstract (PGO and IVE)

·       English should be accurately revised

·       Lines 32-34: are authors sure about this definition of a probiotic?

·       Line 38-39: could authors give a definition of first and second-generation probiotics?

·       Did IVE mean Illicium Verum Extract (line 41), or did it represent the vegetable (as written in line 42)?

·       The introduction could be enriched with further information, for example about the bactericidal role of IVE, protecting the role of probiotics on intestinal microflora, previous experimentation on piglets, and why the use of this animal model…

·       Were piglets (Duroc × Landrace × Yorkshire) randomly assigned to the treatments?

·       Table 1 was not mentioned in the text

·       Table 3 is not clear as presented. Authors should write each acronym in the caption (e.g. BV was never defined in the manuscript)

·       Figure 5: authors should write in the caption what each figure (A1, A2…) represents.

·       Lines 185 – 193: the paragraph is not clearly presented, and it is difficult to understand

·       Line 201: Western blot

·       Line 27a: Studies

·       What are the possible future implications of these findings?

Author Response

Reply to Editor and reviewers

On behalf of my co-authors, we sincerely thank you for giving us an opportunity to revise our manuscript, and deeply appreciate editor and reviewers for their positive and constructive comments and suggestions on our manuscript entitled “The combination of plant extracts and probiotics improved jejunal barrier and absorption capacity of weaned piglets”.

We have checked the revised manuscript carefully, and look forward to hear from you soon. If you have any questions, please don’t hesitate to let us know. Appended to this letter is our point-by-point response to the comments raised by the reviewers. The comments are reproduced and our responses are given directly afterward in red color.

Reviewer #1: Comments and Suggestions for Authors

  1. Abbreviations should be defined starting from the abstract (PGO and IVE).

- Thanks for the your kind reminder. The “PGO and IVE” have been defined in the section of abstract.

  1. English should be accurately revised

- We are grateful for the suggestion. We have tried our best to revise our manuscript according to the comments. The revised manuscript also has been completed by a colleague proficient in English (Yang C.W. Guelph Research and Development Center, Agriculture and Agri-Food Canada (AAFC), 93 stone road west, Guelph, Ontario, Canada). The changes have been marked in our manuscript, if you have any questions, please don’t hesitate to let us know. 

  1. Lines 32-34: are authors sure about this definition of a probiotic?

- We thank the reviewer for the very important comment. The sentence “Probiotics are microbial formulation made from normal microorganisms or substances that promote microbial growth. In other words, all preparations that can promote the growth of normal microbiota and inhibit the growth of pathogenic bacteria are named "probiotics"" was changed to “Probiotics are defined as living microorganisms that have beneficial dffects on host health when ingested in sufficient quantities”. This definition comes from Hill et al (Hill C, Guarner F, Reid G, Gibson GR, Merenstein DJ, Pot B, Morelli L, Canani RB, Flint HJ, Salminen S, Calder PC, Sanders ME. The International Scientific Association for Probiotics and Prebiotics consensus statement on the scope and appropriate use of the term probiotic[J]. Nature reviews Gastroenterology & hepatology, 2014, 11(8): 506-514).

  1. Line 38-39: could authors give a definition of first and second-generation probiotics?

- Thanks for your positive comments. “The first generation probiotics are mainly used in the food industry, focusing on the health function of the body. The second-generation probiotics refer to the specific strains with sufficient dose that can help the body recover the healthy phenotype, or can use the existing strains as the carrier of health promoting molecules, so they are also called live biotherapeutics” was added. (OToole PW, Marchesi JR, Hill C. Next-generation probiotics: the spectrum from probiotics to live biotherapeutics[J]. Nature microbiology, 2017, 2(5): 1-6.)

  1. Did IVE mean Illicium Verum Extract (line 41), or did it represent the vegetable (as written in line 42)?

- Thanks for the kind remind. The “IVE” was replaced with “illicium verum” (Line 42).

  1. The introduction could be enriched with further information, for example about the bactericidal role of IVE, protecting the role of probiotics on intestinal microflora, previous experimentation on piglets, and why the use of this animal model…

 - We are grateful for the kind reminder. The sentence “The ways of probiotics regulating animal intestinal health mainly come from two aspects. Dominant flora theory: first, probiotics adhere to intestinal mucosal epithelial cells by competing for nutrients and parasitic sites, and inhibit the attachment and growth of harmful bacteria. Biological oxygen capture theory: secondly, when the spores of probiotics enter the intestinal tract, they accelerate the consumption of oxygen in the environment, ensure the anaerobic environment and the growth of dominant bacteria, and ultimately improve the production performance of the host [12-14].” and “Studies have shown that Ive can exert its antibacterial function by inhibiting the formation of biofilm and reducing bacterial adhesion [19].” were added in the section of introduction.

  1. Were piglets (Duroc×Landrace×Yorkshire) randomly assigned to the treatments?

- Thanks for your kind comments. In our study, thirty two piglets were selected and randomly divided into four treatments. Relevant details have been supplemented in our manuscript.

  1. Table 1 was not mentioned in the text

- Thanks for your reminder. The “(Table 1)” was added in the section of “IVE and probiotics supplemented diet”.

  1. Table 3 is not clear as presented. Authors should write each acronym in the caption (e.g. BV was never defined in the manuscript)

- Thanks for your reminder. The sentence “2DM, dry matter; OM, organic matter; EE, ether extract; CP, crud protein; BV, biological value; NPU, net protein utilization.” was added in the caption of Table 3.

  1. Figure 5: authors should write in the caption what each figure (A1, A2…) represents.

- Thanks for the reviewer’s evaluation. The sentence “Treatments were basal diet only (IVE-PGO-, A), basal diet + 1000 mg/kg PGO (IVE-PGO+, B), basal diet + 500 mg/kg IVE (IVE+PGO-, C), and the basal diet + 500 mg/kg IVE + 1000 mg/kg PGO (IVE+PGO+, D), respectively.” was added in the caption of Figure 1-3.

  1. Lines 185-193: the paragraph is not clearly presented, and it is difficult to understand

- Thanks for your kind comments. The sentence “Immune positive distribution of SGLT1 and ZO-1 in jejunum was shown that the expression of SGLT1 in IVE-PGO+, IVE+PGO- and IVE+PGO+ groups was mainly distributed in the middle of jejunal villi, and the positive reaction gradually increased from the top to the middle. In addition, the positive reactions was also found in the chylous duct of villi, which were scattered. The positive reaction of IVE-PGO- villi was stronger at the bottom of villi, but weaker at the upper and middle villi. The expression sites of ZO-1 were mainly distributed in the microvilli of jejunum and the wall of villous chylous duct. What's more, the positive spots on the wall of villous chylous duct were mostly beaded.” was changed to “The immunopositive distribution of SGLT1 and ZO-1 in jejunum indicated that the expression of SGLT1 in IVE-PGO+, IVE+PGO- and IVE+PGO+ groups was mainly distributed in the middle of jejunum villi, and gradually increased from the top to the middle. In addition, there were positive reactions in the villous chylous ducts. In the IVE-PGO- group, the positive reaction was strongest at the base of villi, but weaker at the upper and middle villi. The expression sites of ZO-1 are mainly distributed in the microvilli of jejunum and the wall of chylous villi, and the positive spots on the wall of chylous villi are mostly beaded”.

  1. Line 201: Western blot

- Thanks, “wetern-blot” was replaced with “western-blot”.

  1. Line 27a: Studies

- Thanks, “Sdudies” was replaced with “studies”.

  1. What are the possible future implications of these findings?

- We are grateful for the comments. Under the global trend of limiting the use of antibiotics, our research provides a new idea for the selection of antibiotic substitutes. At present, among the most alternative antibiotics, the first generation probiotics are widely used, but the effect is not ideal when used alone. In our scheme, we use the characteristics of plant extracts to inhibit harmful bacteria, select probiotic strains that can coexist with specific plant extracts, and use them in combination, which can not only eliminate harmful bacteria, but also help the host to establish a healthy intestinal flora. Therefore, we believe that the combination of plant extracts and probiotics may be an important strategy to improve animal health before the advent of antibiotic substitutes with absolute advantages. Relevant descriptions have been added in the section of conclusion.

Reviewer 2 Report

1. Please explain synergical effects of probiotic and plant extracts.

2. How to prove the survial of probiotics?

3. Result and disscussion must improve extensive.

Author Response

Reply to Editor and reviewers

On behalf of my co-authors, we sincerely thank you for giving us an opportunity to revise our manuscript, and deeply appreciate editor and reviewers for their positive and constructive comments and suggestions on our manuscript entitled “The combination of plant extracts and probiotics improved jejunal barrier and absorption capacity of weaned piglets”.

We have checked the revised manuscript carefully, and look forward to hear from you soon. If you have any questions, please don’t hesitate to let us know. Appended to this letter is our point-by-point response to the comments raised by the reviewers. The comments are reproduced and our responses are given directly afterward in red color.

Reviewer #2: Comments and Suggestions for Authors

  1. Please explain synergical effects of probiotic and plant extracts.

- Thanks for your kind reminder. The relevant description “In our strategy, Ive reduces the abundance of harmful bacteria by inhibiting the formation of bacterial biofilm. In addition, when Ive is used in combination with probiotics, probiotics and harmful bacteria competitively adhere to intestinal mucosal epithelial cells, compete with pathogenic microorganisms for nutrients and ecological sites, so as to enhance the biological barrier function of animal intestines and help the host establish a healthy intestinal flora, so as to improve the digestibility and growth performance of nutrients.” was added in the section of “Discussion”. Under the global trend of limiting the use of antibiotics, our research provides a new idea for the selection of antibiotic substitutes. At present, among the most alternative antibiotics, the first generation probiotics are widely used, but the effect is not ideal when used alone. In our scheme, we use the characteristics of plant extracts to inhibit harmful bacteria, select probiotic strains that can coexist with specific plant extracts, and use them in combination, which can not only eliminate harmful bacteria, but also help the host to establish a healthy intestinal flora. Therefore, we believe that the combination of plant extracts and probiotics may be an important strategy to improve animal health before the advent of antibiotic substitutes with absolute advantages.

  1. How to prove the survial of probiotics?

- We are grateful for the kind comment. We have realized that the problem you pointed out is caused by the imperfection of our experimental design. As we all know, the difficult colonization of probiotics in the host intestinal tract is a huge obstacle to the promotion and application of the first generation probiotics. Research shows that in the use of most probiotics, probiotics only act as a passing bacterium. However, studies have also shown that probiotics do not necessarily play a role only when they are alive. Extracellular enzymes and ester peptides produced by probiotics when they are alive also play an important role in improving host health. However, we did carry out 16S rRNA sequencing analysis on animal feces, and found that the abundance of some beneficial bacteria such as Christensen increased significantly, but we did not find that the abundance of the genus of probiotics we invested increased. Therefore, we speculate that the probiotics used under the test conditions may play a prebiotic role through its secondary metabolites. As for the detailed mechanism, we need to further study. We believe that your suggestions are the focus of our next research. Thank you again for your valuable suggestions. The relevant description was added in the section of “discussion”.

  1. Result and disscussion must improve extensive.

- Thanks for your kind comment. We have rearranged and described the results in the manuscript according to your suggestions. At the same time, we have further discussed the test results under the test conditions on the basis of supplementing the latest literature. Relevant contents have been marked in red in the manuscript. If you have any questions, please don’t hesitate to let us know.

Reviewer 3 Report

.

Comments to the authors

§  L36: rephrase the sentenve, e.g. have beneficial effects on health ….

§  L37: use humans instead of ‘’people’’

§  L47: add appropriate references

§  L82-84: provide the approval number

§  L85: clarify more clearly the groups of the trial

§  L101: provide details of the euthanasia method  

§  L284-286: you should extend the part of conclusions 

§  L288-295: provide appropriate details for Author Contributions

§  L296-299: provide appropriate details of the funding 

Author Response

Reply to Editor and reviewers

On behalf of my co-authors, we sincerely thank you for giving us an opportunity to revise our manuscript, and deeply appreciate editor and reviewers for their positive and constructive comments and suggestions on our manuscript entitled “The combination of plant extracts and probiotics improved jejunal barrier and absorption capacity of weaned piglets”.

We have checked the revised manuscript carefully, and look forward to hear from you soon. If you have any questions, please don’t hesitate to let us know. Appended to this letter is our point-by-point response to the comments raised by the reviewers. The comments are reproduced and our responses are given directly afterward in red color.

Reviewer #3: Comments to the authors

  1. L36: rephrase the sentenve, e.g. have beneficial effects on health….

- Thanks for your comment. The sentence “Probiotics are microbial formulation made from normal microorganisms or substances that promote microbial growth. In other words, all preparations that can promote the growth of normal microbiota and inhibit the growth of pathogenic bacteria are named "probiotics"" was changed to “Probiotics are defined as living microorganisms that have beneficial dffects on host health when ingested in sufficient quantities”.

  1. L37: use humans instead of“people”.

- Thanks for your kind reminder. The “people” was changed to “humans”.

  1. L47: add appropriate references.

- Thanks for the kind reminder. The references “Zhang J, Liu Y, Yang Z, et al. Illicium verum extracts and probiotics with added glucose oxidase promote antioxidant capacity through upregulating hepatic and jejunal Nrf2/Keap1 of weaned piglets[J]. Journal of animal science, 2020, 98(3): skaa077.” and “Holkem A T, Silva M P, Favaro-Trindade C S. Probiotics and plant extracts: a promising synergy and delivery systems[J]. Critical Reviews in Food Science and Nutrition, 2022: 1-19.” were added.

  1. L82-84: provide the approval number.

- Thanks for your reminder. The approval number “(SDAUA-2020-0710)” was added.

  1. L85: clarify more clearly the groups of the trial.

- Thanks for your kind comment. The sentence “Thirty two piglets (Duroc × Landrace × Yorkshire) were selected and randomly divided into four treatments, which included basal diet, basal diet + 1000 mg/kg PGO, basal diet + 500 mg/kg IVE, and basal diet + 500 mg/kg IVE + 1000 mg/kg PGO.” was changed to “Thirty two piglets (Duroc × Landrace × Yorkshire) were selected and randomly divided into four treatments. Treatment 1 was fed with basal diet (IVE-PGO-), and treatments 2, 3 and 4 were fed the basal diet supplemented with 1000 mg/kg PGO (IVE-PGO+), 500 mg/kg IVE (IVE+PGO-) and 500 mg/kg IVE+1000 mg/kg PGO (IVE+PGO+), respectively.”.

  1. L101: provide details of the euthanasia method.

- Thanks for your kind reminder. “(slaughtered after being fainted by electric shock at the neck)” was added.

  1. L284-286: you should extend the part of conclusions

- We are grateful for the comment. “Under the global trend of limiting the use of antibiotics, our research provides a new idea for the selection of antibiotic substitutes. At present, among the most alternative antibiotics, the first generation probiotics are widely used, but the effect is not ideal when used alone. It is suggested that the the combination of plant extracts and probiotics is an vital strategy to improve animal health before the advent of antibiotic substitutes with absolute advantages.” was added in the section of conclusions.

  1. L288-295: provide appropriate details for Author Contributions

- Thanks for the reminder. We have supplemented the author's contribution according to your suggestions in the section of “Author Contributions”.

  1. L296-299: provide appropriate details of the funding

- Thanks for the reminder. We have supplemented the details of funding according to your suggestions in the section of “Funding”.

Round 2

Reviewer 2 Report

should accept in current version.

Author Response

On behalf of my co-authors, we sincerely thank you for giving us an opportunity to accept our manuscript, and deeply appreciate editor and reviewers for their positive and constructive comments and suggestions on our manuscript entitled “The combination of plant extracts and probiotics improved jejunal barrier and absorption capacity of weaned piglets”.

This manuscript is a resubmission of an earlier submission. The following is a list of the peer review reports and author responses from that submission.